# Leveraging Structured Information for Explainable Multi-hop Question Answering and Reasoning

**Ruosen Li** and **Xinya Du**
Department of Computer Science
University of Texas at Dallas
ruosen.li@utdallas.edu xinya.du@utdallas.edu

## Abstract

Neural models, including large language models (LLMs), achieve superior performance on multi-hop question-answering. To elicit reasoning capabilities from LLMs, recent works propose using the chain-of-thought (CoT) mechanism to generate both the reasoning chain and the answer, which enhances the model's capabilities in conducting multi-hop reasoning. However, several challenges still remain: such as struggling with inaccurate reasoning, hallucinations, and lack of interpretability. On the other hand, information extraction (IE) identifies entities, relations, and events grounded to the text. The extracted structured information can be easily interpreted by humans and machines (Grishman, 2019). In this work, we investigate constructing and leveraging extracted semantic structures (graphs) for multi-hop question answering, especially the reasoning process. Empirical results and human evaluations show that our framework: generates more faithful reasoning chains and substantially improves the QA performance on two benchmark datasets. Moreover, the extracted structures themselves naturally provide grounded explanations that are preferred by humans, as compared to the generated reasoning chains and saliency-based explanations.[1]

## 1 Introduction

Multi-hop question answering (QA) involves answering questions that require reasoning over multiple pieces of information, often spread across different parts of a document or multiple documents. It looks like "hopping" over various facts to arrive at a conclusion or answer. In multi-hop question answering, the system needs to understand the context, maintain the sequence of information, and utilize this combined information to generate a correct and comprehensive answer. Traditionally, researchers (Qiu et al., 2019; Tu et al., 2019; Fang

[1]Code is available at https://github.com/bcdnlp/Structure-QA.

et al., 2020) have applied graph neural networks (GNN) to this task. In recent years, with the growing capabilities of large language models (LLMs), several works propose using prompting to address this task in a few- or zero-shot way (Wei et al., 2022; Ho et al., 2023).

LLMs have achieved superior performance in reasoning-based question-answering tasks. These models have the potential to "reason" and connect multiple pieces of information to generate the answers. However, they still struggle with complex multi-hop questions; and the end-to-end generation nature makes their answering process not explainable. To elicit the reasoning capabilities of LLMs, recent research introduced the chain-of-thought (CoT) mechanism and its variants (Wei et al., 2022; Wang et al., 2023). The CoT mechanism enables models to not only generates better answers but the reasoning chain (process) in natural language, improving the LLM's performance in conducting multi-hop reasoning.

Despite the progress made, several challenges persist under this paradigm, One is the neural models' limitations in tackling compositional problems that require strict multi-hop reasoning to derive correct answers (Dziri et al., 2023). CoT-based methods still conduct generations in an end-to-end way, which is not a strict "symbolic derivation" and often causes inaccurate reasoning (e.g., generating answers that don't exist or two-hops away). Instead, we propose a multi-step approach (Figure 1) to tackle this problem: firstly, constructing the semantic graph structures (SG) with information extraction (IE) and then leveraging this symbolic information (including entities and semantic relations) for strictly guiding the model's reasoning process.

The second challenge is that, although the current model-generated reasoning chain provides explanations, they are only surface-level interpretations, and there is no guarantee that they are *fully*

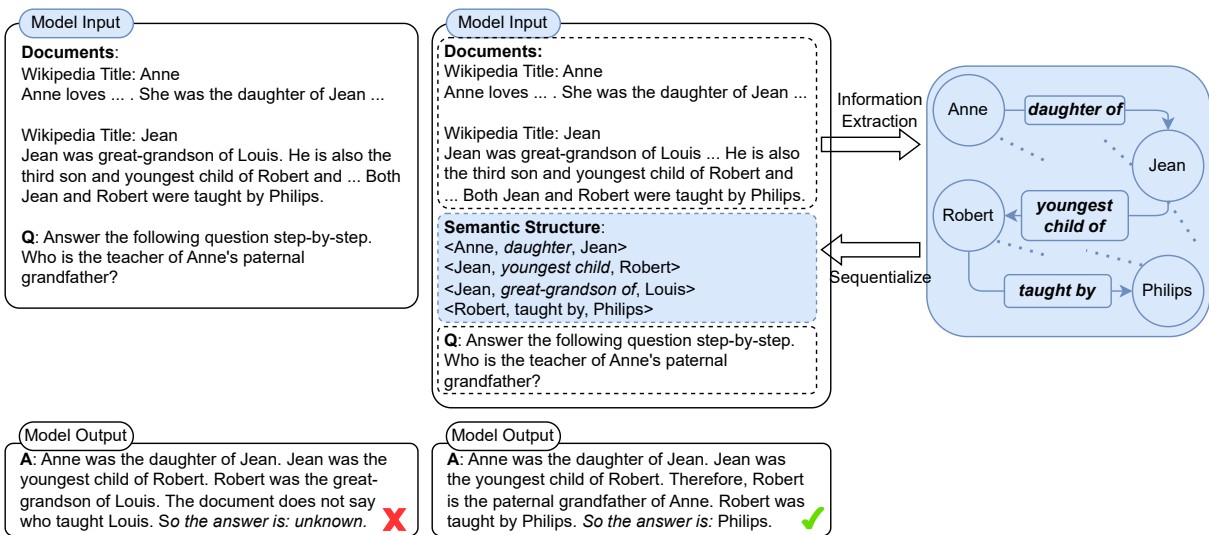

Figure 1: We first extract the semantic graph and add it to the model input prompt. Then they are fed into the LLM to generate the reasoning chain and answer. The semantic structures help improve question answering.

*grounded* to context (Narang et al., 2020). For example, the model could make up a reasonable-sounding explanation that is not faithfulness for its decision-making (hallucinations). While our extracted graphs from the IE step naturally provide explanations grounded to the given context (Figure 4), which are overwhelmingly preferred by human evaluators, compared to saliency- and NLG-based explanations. Plus, we also find that by leveraging our extracted SGs, the large language model also generates more faithful reasoning chains (Section 4.3),

To sum up, we make the following **contributions** in this work: (1) We propose creating and utilizing semantic graphs to enhance models' capability in multi-hop reasoning and achieve better performance on the task; (2) We demonstrate that leveraging semantic graphs in the prompting process help models generate higher-quality and more faithful reasoning chains; (3) We show that semantic graphs-based explanations are grounded to context and help improve the interpretability of the answer-predicting process, which human evaluators prefer.

## 2 Related Work

**Multi-hop Question Answering** Traditional approaches to solving the multi-hop QA problem can be mainly categorized as question decomposition (Perez et al. (2020); Fu et al. (2021); Sun et al. (2021); Deng et al. (2022)), graph-based method (Qiu et al. (2019); Fang et al. (2020)), iterative

method (Asai et al. (2020); Qi et al. (2021)), and miscellaneous techniques (Chen et al. (2020); Ho et al. (2023)).

In these various directions, the graph-based method is a branch that solves the problem by building graphs and inferring on graph neural networks (GNN) to predict answers. Graphs can be categorized as entity-only graphs (Qiu et al. (2019); De Cao et al. (2019)) and multi-level node graphs (Tu et al. (2019); Ye et al. (2019); Fang et al. (2020)). Nodes in those graphs represent entities, sentences, paragraphs, and documents. Edges connect entities and include no semantic information. In our method, we build semantic graphs based on the definition in Section 3.2. Compared to previous work graph building, our constructed graphs include key semantic relations on edges, which provides important fine-grained information.

In addition, under our prompting-based QA setting, we sequentialize the graphs by concatenating the triples, which simplifies the process of encoding the semantic graphs. Specifically, our approach does not require the fine-tuning process since we apply the prompt-based method over LLMs, which simplifies the reasoning process.

This generative format is closely related and concurrent to the prompting-based method that is recently applied to the multi-hop QA task (Trivedi et al. (2022); Khot et al. (2023); Yoran et al. (2023)). We introduce the details and their variance in the paragraph below.

**Prompting and Reasoning** The Chain-of-Thought (CoT) prompting strategy (Wei et al., 2022), as a variation of LLM few-shot prompting, significantly improves the reasoning ability of LLMs. Compared to traditional few-shot prompting, it turns complex implicit reasoning chains into explicit long natural language text and adds it to the demonstrations, which helps LLMs conduct better reasoning. Kojima et al. (2023) introduce zero-shot-CoT by using a simpler prompt ("Let's think step-by-step") to instruct LLMs. Wang et al. (2023) introduce the self-consistency strategy to improve the performance of the CoT strategy.

When applying prompting strategy to the multi-hop QA task, Trivedi et al. (2022) introduces an iterative information retrieval method to provide more relevant context. Khot et al. (2023) decomposes prompts to simple solvable tasks. In our approach, we leverage semantic graphs extracted from documents, which guide reasoning processes and help accurately find the answer node. Those graphs are extracted from context, sequentialized into text form, and added back into context.

Generating faithful reasoning chains is difficult with the vanilla CoT strategy, as generated natural language contents may contain irrelevant information and cause hallucinations. Lyu et al. (2023) try to solve it by translating natural languages to symbolic languages and deducing results by deterministic external solvers. But, the process of translation itself remains opaque and cannot guarantee the prevention of hallucinations.

Instead of directly generating symbolic presentations from contexts, we introduce semantic graphs to prevent hallucinations and improve interpretability. Since graphs are extracted from contexts, they are strictly grounded to context. Edges in the graphs can be seen as potential reasoning steps in the whole reasoning process.

## 3 Methodology

In this section, we first give a formal introduction to the multi-hop QA task and two different settings we tackle the problem. We then describe our technique for extracting semantic graphs, which will be used in our framework. Third, we introduce our overall and specific prompting for generating the reasoning chains and final answers.

### 3.1 Task and Settings

Given a question and candidate paragraphs, we aim to obtain an answer that involves multi-hop reasoning. We tackle the task with in-context learning from two different settings: (1) generate the answer as well as the explanations. – one representative method is chain-of-thought reasoning (Wei et al., 2022). We name it "*CoT* setting" here. – it is our default setting and of more importance since we focus on studying the faithfulness and interpretability of explanations; (2) directly generate the answer from the context and question ("*fewshot* setting").

### 3.2 Entity and Relation Extraction with Prompting

Drawing insights from the literature on information extraction, we propose two paradigms on entity and relation extractions: (1) treating them as separate tasks and conducting multi-step prompting for extracting entities first, and then relations in between them (Zelenko et al., 2003); (2) joint extraction (Li et al., 2013; Miwa and Bansal, 2016) of the entity and relations, thus building the structured semantic graph in an end-to-end way.

For **entity** (node), we define it as a short text span/sequence that clearly describes an object/event/truth in a sentence. – This formulation has broader coverage than traditional named entity recognition (NER) based on a fixed schema. For example, "<Windermere, is popular for, its proximity to the lake>" is an important triple in the semantic graph extracted from "Tourism is popular in Windermere mainly for its proximity to the lake." Traditional NER tools can only extract "Tourism", "Windermere", and "lake", real-world objects, but not "its proximity to the lake", a description of truth. We find in the prelim study that using traditional NER doesn't provide improvements, mainly because they are uninformative. For **relations** (edges), we borrow ideas from Open IE (Fader et al., 2011). Traditional relation extraction classifies relations into sets of categories, for example, OWNERSHIP, LOCATED, FAMILY)[2]. Thus, the number of closed relations is limited. They can hardly be used to present complex relations between entities that are defined above. OpenIE extracts any strings in the sentence that can describe relations between two entities. For our case, we extract the document-level relations described in

---

[2]In total, 17 relations are used in ACE relation extraction evaluations; and Wikidata has 11,008 relations.

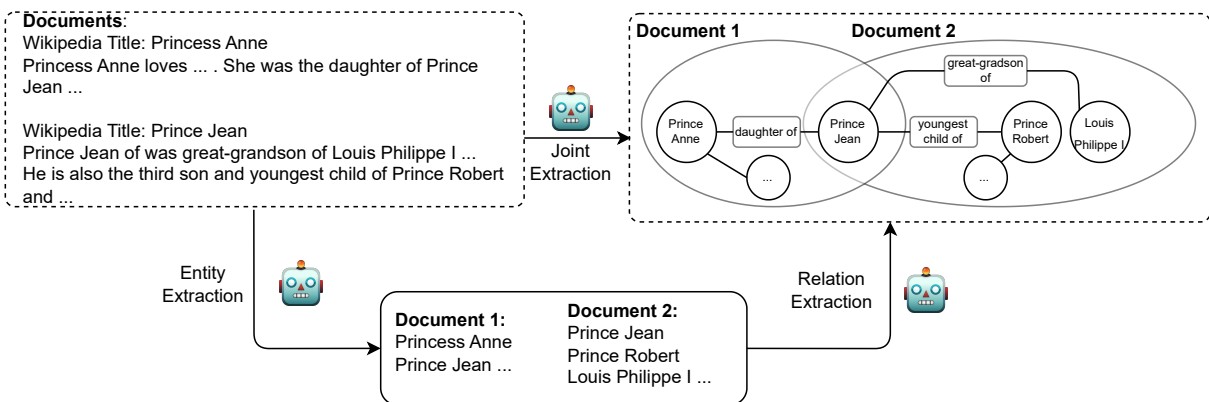

Figure 2: We propose to use single-step prompting for jointly extracting entity-relation graphs; as well as using multi-step prompting for extracting the entities and relations separately. The extracted semantic graph captures the inter-document and intra-document dependencies between entities.

natural language in the text.

Finally, to illustrate in the **semantic graph**, we represent the entities as nodes and semantic relations as edges (e.g., Figure 2). For example, the node "Princess Anne" is connected to the node "Prince Jean" via the edge "daughter of".

**Entity Extraction** The input we use as the prompt for the entity extraction step basically consists of one document, including a Wikipedia paragraph. It starts with the word "`Document:`". Then, it is followed by a paragraph with a title. Finally, we add to the prompt `Entities:`, asking it to start extracting a list of entities:

```
Document:
Wikipedia Title:  <Page Title>
<Paragraph Text>
Entities:
```

The output to be generated by the LLM is supposed to be all relevant entities listed line-by-line. Overall, it looks like:

```
            <Entity 1>
            <Entity 2>
            ...
            <Entity k>
```

To better instruct the model following the output format, we utilize few-shot in-context learning (ICL) to prompt the LLMs (Brown et al., 2020). Specifically, we add four document-entities pair as the demonstration.

**Semantic Relation Extraction** To extract the semantic relations between entities, we propose two variations, one uses the multi-step extraction idea, and the second conduct joint extraction. Specifically, for multi-step extraction, the model extracts the entities first (as mentioned above), then append them to the prompt of relation extraction, and finally the prompt ends with `Graph:`,

```
Document:
Wikipedia Title:  <Page Title>
<Paragraph Text>
<Entity 1>\n ... \n<Entity k>
Graph:
```

The output to be generated would look like this:

```
(<Entity a>, <R 1>, <Entity b>)
(<Entity b>, <R 2>, <Entity c>)
...
```

We also conduct a few-shot ICL. In this graph structure, not all pairs of entities have relations. We extract relations for pairs of entities if it exists.

**Joint Extraction of Entity and Relation Triples** Different from the multi-step of extracting semantic graphs above, we also try direct prompting. Namely, we skip the entity generation process and directly generate the (subject, relation, object) triples with prompting. Basically, we directly add `Graph:` by the end of the paragraph context.

```
Document:
Wikipedia Title:  <Page Title>
<Paragraph Text>
Graph:
```

We name the semantic graph generated through multi-step prompting "SG-Multi"; the semantic graph generated through the single step (joint extraction) "SG-One".

To achieve a comprehensive understanding of how the graph structure affects the QA and reasoning process. We also build a fully connected version of the graph based on all the entities extracted in Step 1. More specifically, suppose there are $k$ extracted entities, there would be $k^2$ tuples, the graph would look like:

```
(<Entity 1>, <Entity 2>)
(<Entity 1>, <Entity 3>)
...
(<Entity k-1>, <Entity k>)
```

Since the fully connected graph does not leverage semantic information, we call it "G-Full".

### 3.3 The Overall Prompt for the Question Answering Task

For each question, we have multiple associated passages as context. For each passage, a semantic graph is generated as described above. Then, all the graphs are combined together to form the overall prompt template. Thus, the overall prompt template includes three components: documents, an extracted graph, and a question prompt.

The input part consists of multiple evidence documents followed by their corresponding graphs. Then all of these structures are concatenated. Since our default setting is chain-of-thought, as suggested by Trivedi et al. (2022), we add "Answer the following question by reasoning step-by-step" ahead of the real question.

```
Documents:
Wikipedia Title:  <Page Title 1>
<Paragraph Text 1>
<graph 1>···
Wikipedia Title:  <Page Title n>
<Paragraph Text n>
<graph n>
Q: "Answer the question by
reasoning step-by-step" <question>
A:
```

Then the output part would be like below:

```
<Reasoning Steps>:  <answer>
```

For the example question in Figure 1, the reasoning steps would be "Princess Anne was the daughter of Prince Jean. Prince Jean was the youngest child of Prince Robert. Prince Robert is the paternal grandfather of Princess Anne." One interesting difference from our reasoning chain (as specified in the demonstration) to the default CoT (Wei et al., 2022) tasks' is that, under the multi-hop setting, we can use a specific format for the multi-hop question reasoning process. More specifically, we define a reasoning chain as a list of short sentences in the correct logical order, and each sentence only represents a relation between two entities. The set of sentences naturally leads to the answer entity. Under this format, the reasoning chain can be more consistent, facilitating more fair and automatic evaluations.

Finally, we post-process the generated `<answer>` by applying a regex expression to extract answers from the `<answer>` after it is generated – they might have a slightly different format or length to the real/gold answer. For the simple few-shot setting (non-CoT), we remove "`Answer the question by ...`" and "`Reasoning Steps`" from the few-shot examples/demonstrations.

## 4 Experiments

### 4.1 Datasets and Configuration

We evaluate our model on two multi-hop question-answering datasets: HotpotQA (Yang et al., 2018) and 2WikiMultiHopQA (Ho et al., 2020). Both datasets have ten candidate paragraphs for each question and originally have supporting facts. We use the golden paragraphs as context for all questions. Following Trivedi et al. (2022), for each dataset, we randomly sample 100 questions from the original development set and use it as our development set for tuning hyperparameters. We randomly sample another 500 questions from the development set as our test set.

We use the model GPT-3.5[3] for the entity, relation, and graph extractions, as well as the final QA task. For entity and graph extractions, we add four demonstrations. For the question answering, we add two demonstrations. We set the maximum number of generation tokens for the entity and graph extraction process to be 300. There is no number limitation generating the reasoning chain and

---

[3]https://platform.openai.com/docs/models/gpt-3-5

| | EM | F1 | Precision | Recall | EM | F1 | Precision | Recall |
|---|---|---|---|---|---|---|---|---|
| Methods | | HotpotQA | | | | 2WikiMultiHopQA | | |
| Base | 0.59 | 0.733 | 0.745 | 0.779 | 0.43 | 0.532 | 0.501 | 0.724 |
| G-Full prompt | 0.62 | 0.763 | 0.777 | 0.813 | 0.46 | 0.565 | 0.534 | 0.757 |
| SG-Multi prompt | 0.64 | **0.785** | 0.791 | **0.849** | 0.49 | **0.601** | 0.573 | **0.772** |
| SG-One prompt | 0.61 | 0.749 | 0.741 | 0.823 | 0.47 | 0.579 | 0.563 | 0.748 |

Table 1: CoT setting results. SG-Multi and SG-One generally outperforms G-Full and the base prompt. This demonstrates the importance of explicitly extract and encoding the semantic relations.

| | F1 | Precision | Recall | F1 | Precision | Recall |
|---|---|---|---|---|---|---|
| Methods | | CoT Setting | | | Few-shot Setting | |
| Base | 0.716 | 0.729 | 0.830 | 0.6280 | 0.5970 | 0.7821 |
| G-Full prompt | 0.714 | 0.732 | 0.805 | 0.6065 | 0.5910 | 0.7802 |
| SG-Multi prompt | 0.728 | 0.731 | **0.872** | 0.5925 | 0.5596 | **0.8205** |

Table 2: Few-shot Setting Recall Results on HotpotQA.

answer span. But, the generation stops when it changes line. The "temperature" parameter is set to 0 when generating all predictions.

Regarding metrics, we use exact match (EM), F1, Precision, and Recall. We run the official evaluation scripts of each dataset to get the output scores.

## 4.2 Evaluation on the QA task

We first run experiments on both datasets under the CoT setting. We report results in Table 1.

On HotpotQA, we observe that all prompting-based methods achieve decent performance, and by adding graph structures, we see an improvement across all four metrics. Especially, the "SG-Multi" prompt improves the recall by 4%. When the model generates reasoning chains and answers, SG enables the LLM to "hop through" the structured graph and context instead of inferring relations between entities purely from context. The reason that the recall score goes down (from 0.83 to 0.81) by using G-Full is that it sometimes provides spurious implicit related information (i.e., all entity pairs). Then the sequentialized graph would be too long and doesn't fit into the LLMs' input length constraint. Then it is harder for the model to capture an entire picture of the relations needed to answer the question.

On 2WikiMultiHopQA, we see a similar trend of improvements. The results of exact match, F1, and precision increase substantially after adding the graphs. They are improved by 9%, 6%, and 6% on average as compared to the base prompt. Recall

goes up in both the G-Full, SG-Multi, and SG-One settings. After manual analysis and comparisons to the Hotpot, we find that the graphs are smaller and of higher quality, which leads to a more faithful chain and final results. This will be discussed further in Section 4.3.

Further, we also present the recall performances for the few-shot setting (w/o CoT) in Table 2. We see recall improved by adding the structures, especially using SG-Multi for both HotPotQA and 2WikiHop. We will mention below why recall is a better metric for the multi-hop QA task when comparing prompting-based methods.

**Recall, precision and other metrics** We argue that recall is a better metric for this task among the four metrics under the generative QA setting, where LLMs are used to *generate* the reasoning chains and answers.

Different from extractive QA setting based on fine-tuned discriminative models: (1) generated answer's length varies, both wider (lower precision) or shorter (lower recall) answers can be correct; (2) LLMs sometimes generate tokens that are not shown/grounded to the original documents. – They can be in a longer-form (Fan et al., 2019) that includes tokens outside golden answers but don't exactly match (EM). From the human's judgment, most of those predicted answers may be correct. For example, the gold answer to the question "Where was the father of Knut Hedemann born?" is "Stange". But "Stange, Norway" is also correct, which provides extra information. In this specific

example, the recall metric is more lenient and provides the same score. But the precision is punished. Without fine-tuning, text generation is less controllable than text span extraction. Thus, the precision metric is less suitable for the generated answers. Since F1 is affected by precision, it also punished slightly longer but correct answers.

|       | EM   | F1   | Prec | Recall |
|-------|------|------|------|--------|
| $\rho$ | 0.51 | 0.77 | 0.76 | 0.88   |
| $\tau$ | 0.51 | 0.69 | 0.68 | 0.87   |

Table 3: Correlations of metrics and human judgments.

To empirically verify, we conduct a brief manual study. Firstly, we manually check if generated answers are correct/acceptable under human judgment. If we believe an answer is correct, we assign 1 as its score; otherwise, 0. Then we calculate both Spearman ($\rho$) and Kendall-Tau ($\tau$) correlation to scores provided by each metric. As shown in Table 3, recall corresponds more to human evaluation.

Apart from recall, according to Table 1, we find that **precision** gets slightly increased (with hurting recall) by adding the SG-Multi (or SG-One), especially on the 2WikiMultiHopQA. As discussed previously, precision is highly influenced by the length of predicted answers. By adding graphs, the QA and reasoning process is more likely to refer to entity-relation triples in the graphs rather than starting from scratch from the long documents. According to our entity definition above and answer format, they are brief and informative. Thus, the final predicted answers are sometimes more accurate, which helps improve the precision and F1 score.

### 4.3 Evaluation on the Reasoning Chain

Apart from evaluating the quality of answers, it is also important to evaluate free-form reasoning chains under our CoT setting. We sample 100 questions in the 2WikiMultiHopQA dataset and manually annotate their reasoning chains (explanations) according to the format defined in Section 3.3. For the predicted reasoning chains, we conduct both human and automatic evaluations. We make the human correctness judgments based solely on factuality and hallucinations/faithfulness[4]. They are also the two most important considerations in Golovneva et al. (2022). Human judges decide whether one reasoning process is "correct/good"

---

[4]Coherence and grammar mistakes rarely occur.

based on the two requirements. During our study, our two annotators fully agree.

Although reliable, human evaluation is expensive. Motivated by this, recent work has also investigated automatic metrics for NL explanations evaluation, including word overlap or embedding based similarly with human written explanations (Clinciu et al., 2021). We consider two candidate automatic metrics: **ROUGE** measures n-gram overlap between generated and reference texts based on recall of important content. **BERTScore** (Zhang et al.) uses BERT embeddings to capture contextual information and computes a similarity score based on cosine similarity between generated and reference text embeddings. We first check the correlations between automatic metrics and human evaluations on whether the reasoning process is faithful or not. For each metric, they provide a score between 0 to 1 (with 1 as the perfect match). We calculate Spearman ($\rho$) and Kendall-Tau ($\tau$) correlations. We find that ROUGE is a slightly better metric for auto-eval (with $\rho$ being around 0.32 and $\tau$ around 0.27), as compared to BERTScore (with $\rho$ being around 0.24 and $\tau$ around 0.20). Overall, this shows a moderate agreement with human evaluations.

On our sampled questions, Table 4 shows the performance of generated reasoning chains by each method based on corresponding prompts. We show both human and automatic evaluation scores. We see that based on strong LLMs, all three prompting-based methods achieve good scores, but the prompt augmented with rich semantic structured (SG-Multi) is of better quality. Below we look into examples and provide a more fine-grained evaluation and analysis.

**More manual evaluation and analysis** We sample and compare 100 pairs of reasoning chains generated by three prompting-based methods on the 2WikiMultiHopQA dataset. There are mainly three kinds of benefits/differences that our SG-augmented prompts bring,

- Improvement of the faithfulness of reasoning chains. In Figure 3, there is a comparing example on the difference. Both reasoning chain starts with truthful facts (around two sentences). Later, the base prompt-based method generates Upper sing, which is either a city ("Grafelfing") or a state name ("Upper Bavaria"), which causes the error of the final answer as well.

- Help the model navigate through and conduct

| method | Human | ROUGE-1 | ROUGE-2 | ROUGE-L |
|---|---|---|---|---|
| Base Prompt | 0.81 | 0.680 | 0.520 | 0.614 |
| G-Full prompt | 0.87 | 0.687 | 0.536 | 0.624 |
| SG-Multi prompt | 0.88 | 0.689 | 0.543 | 0.628 |

Table 4: Evaluations of Generated Reasoning Chains.

Wikipedia Title: Marcel (song)
"Marcel" was the German entry in the Eurovision Song Contest 1963, performed in German by Heidi Brühl ...
Wikipedia Title: Heidi Brühl
Heidi Rosemarie Brühl (30 January 1942 in Gräfelfing, Upper Bavaria – 8 June 1991 in Starnberg) was a German singer and actress [...]

**Graph**:
("Marcel", performed by, Heidi Brühl)
(Heidi Rosemarie Brühl, was born in, Gräfelfing)
(Gräfelfing, located in, Upper Bavaria) ...

**Question**:
What is the place of birth of the performer of song Marcel (Song)?

**Reasoning chian (base prompt)**:
The performer of the song "Marcel" is Heidi Brühl. According to the Wikipedia page about Heidi Brühl, she was born in Gräfelfing, Upper Sing in Germany. So the answer is: Germany Sing.

**Reasoning Chain (SG-One prompt)**:
The performer of the song Marcel is Heidi Brühl. Heidi Rosemarie Brühl was born on January 30, 1942, in Gräfelfing, which is located in Upper Bavaria. So the answer is: Gräfelfing, Upper Bavaria.

Figure 3: Reasoning chain and answer generated by our framework are more faithful. The base prompt leads to "Germany Sing" which didn't appear in context.

more accurate reasoning (and know when to stop). For the example in Figure 4, in the question "Who is the paternal *grandfather* of Princess Anne of Orléans?", the reasoning chain generated by the based prompt has an "over-reasoning" phenomenon (conducting more hops of reasoning). – It generates a name that is the *great-great-grandfather* of Princess Anne of Orléans, "Louis Philippe I". While in the reasoning chain generated by inputting SG-Multi prompt, the reasoning process "stop" at the correct answer (i.e., "Prince Robert, Duke of Chartres").

• When the quality of the extracted semantic graph is low, the reasoning chain's quality will be af-

fected. Most of the time, the poor quality is caused by the incompleteness of the generated graph. Out of the 100 questions, about three examples exhibit such problems. Basically, the extracted SG is incomplete because of the output length limitation of the LLM. Thus the information is lost in the extracted graphs. This prompts the model to output "The graph does not provide information about the question." instead of the reasoning chain.

To summarize the above, we find that generation of reasoning chains relies on SG's quality. With better-quality graphs, both final answers and reasoning chains can be more accurate.

## 5 Further Qualitative Analysis

**Error Analysis** We further investigate errors made by our SG-based framework. There are four major categories: (1) lack of external or commonsense knowledge (e.g. one's uncle is the brother of father), most of the wrong answers are caused by this. (2) poor quality graph (limitation of input length); (3) other problems such as metaphors/paraphrasing (e.g. relations between "rests" and "death") and mathematical knowledge. (4) unanswerable questions based on the given context, which is a problem of dataset creation. Around 5% of questions have this problem.

**More about Explanabiltiy/Interpretability** Our extracted semantic graph *naturally* provides an explanation of the reasoning process. Current work mainly relies on the generated reasoning chain, which is NLG-based and contains hallucinations (Golovneva et al., 2022). Our SG is fully grounded to the input context, as shown in the upper part of Figure 4. The red parts are the extracted nodes, and the green parts denote the relations/edges. Apart from comparing to NLG-based (reasoning chains in our setting) explanations (Narang et al., 2020). We also compare to the prevalent saliency-based method, which obtains explanations that are also grounded to the context

**Documents**:
Wikipedia Title: Princess Anne of Orléans
Princess Anne of Orléans ... She was the daughter of Prince Jean, Duke of Guise and Princess Isabelle of Orléans.

Wikipedia Title: Prince Jean, Duke of Guise
Prince Jean of Orléans, Duke of Guise [...], was the third son and youngest child of Prince Robert, Duke of Chartres (1840 – 1910), grandson of Prince Ferdinand Philippe and great-grandson of Louis Philippe I, King of the French. ...
**Graph**:
(Princess Anne of Orléans, daughter of, Prince Jean)
(Prince Jean, was, grandson of Prince Ferdinand Philippe)
(Prince Jean, was, great-grandson of Louis Philippe I)
[...]
**Question**:
Who is the paternal grandfather of Princess Anne Of Orléans?

**Documents**:
Wikipedia Title: Princess Anne of Orléans
Princess Anne of Orléans ... She was the daughter of Prince Jean, Duke of Guise and Princess Isabelle of Orléans.

Wikipedia Title: Prince Jean, Duke of Guise
Prince [...] youngest child of Prince Robert, Duke of Chartres[...] Philippe and great- grandson of Louis Philippe I, [...]
**Question**:
Who is the paternal grandfather of Princess Anne Of Orléans?

Figure 4: Our extracted triples are highlighted (upper), which naturally provided grounded interpretations. While the saliency-based method provides noninformative ones (bottom). Answers are underlined.

(bottom of Figurer 4). It's mainly done by using attention scores to highlight the most important spans/words in the context (Lei et al., 2016). We conduct rigorous human preference studies. We invite a group of over four volunteers (with at least bachelor's degrees) to make pairwise comparisons of explanations in a blind way. They are only provided with ten questions, the corresponding contexts, and the highlighted words in the context (that models deem as most important for answering questions). Volunteers overwhelmingly vote for our SG-based extractive graphs/explanations as most grounded and informative. Beyonds, results also show a large preference for grounded explanations as compared to the reasoning chains, mainly because they have to verify the factuality/faithfulness of the *generated* explanations.

## 6 Conclusion

We investigate how extracted semantic graphs can contribute to explainable multi-hop question answering. We propose a prompting-based method that leverages the extracted and sequentialized graph into a prompt context. Through comprehensive experiments from multiple aspects, we find that leveraging the SGs substantially improves LLMs' reasoning capabilities in answering multiple-hop questions – with more faithful reasoning chains and better accuracy scores. Meanwhile, human studies show that the extracted graph, which itself is an interpretation that is strictly grounded to context, is preferred. As compared to the NLG-based explanations such as the generated reasoning chain and saliency-based explanations.

## Limitations

- Since our framework adds an extra step of extracting the semantic graphs, the total inference time of the large language model is longer, and the cost is more.

- Sometimes, the extracted semantic graph's quality is unsatisfactory. It's mainly because of the limitation of context window size, which causes the graph to be incomplete. This affects the generation of reasoning chains as well as answering the questions.

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
