# OpenReview forum: "Leveraging Structured Information for Explainable Multi-hop Question Answering and Reasoning"
_EMNLP/2023/Conference — EMNLP 2023 Findings_

### Official Review · Reviewer_3aEg · 2023-07-28

**Soundness:** 3

**Excitement:**

3: Ambivalent: It has merits (e.g., it reports state-of-the-art results, the idea is nice), but there are key weaknesses (e.g., it describes incremental work), and it can significantly benefit from another round of revision. However, I won't object to accepting it if my co-reviewers champion it.

**Paper Topic And Main Contributions:**

This paper leverages structured information like entities and relations to enhance LLM with better CoT capability for solving multi-hop QA tasks. This work prompts the LLM to additionally to extract the entities and their relations in the input documents. By grounding on the extracted structured information, the model can generate higher-quality and more faithful reasoning chains and hence improve the performance of multi-hop QA tasks.
The main contribution lies in the better prompt algorithm and NLP engineering experiments.

**Questions For The Authors:**

1. Since the LLM is grounded to structured information during reasoning, I believe the performance of  IE is very important. Could you test the LLM in terms of IE capability? Such as evaluating some NER or relation extraction benchmarks.
2. In line 87, the explanations generated by your method are grounded to the extracted information. How do you verify this? I did not find any experiments to support this claim. Your method can improve the performance of QA, but it does not mean that the model really grounded to the extracted information.

**Reasons To Accept:**

1. The motivation of this paper is clear. That is, ask the model to pay attention to the structured information from input documents and make faithful reasoning.
2. This method improves the quality of reasoning chains generated by the LLM.

**Reasons To Reject:**

1. Although the methodology presented in this study seems to be interesting, the overall evaluation lacks rigor. The authors conducted experiments on two multi-hop question-answering datasets, each containing a mere 100 questions. Furthermore, the human evaluation was carried out with only 25 questions. Such a limited scope of evaluation is insufficient to derive reliable conclusions.
2. The performance improvement is not significant. Is it worthwhile to consume a lot of computation to process such a long prompt for only a marginal improvement?
3. As indicated in this paper, the primary contribution of this work is the demonstration that LLM grounded to structured information can enhance the reasoning process. Consequently, the performance of LLM concerning Information Extraction (IE) capability holds significant importance. Nevertheless, this article does not present the IE outcomes of LLM, raising concerns regarding the effectiveness of the extracted information in genuinely providing assistance.

**Reproducibility:**

3: Could reproduce the results with some difficulty. The settings of parameters are underspecified or subjectively determined; the training/evaluation data are not widely available.

**Reviewer Confidence:**

3: Pretty sure, but there's a chance I missed something. Although I have a good feel for this area in general, I did not carefully check the paper's details, e.g., the math, experimental design, or novelty.

---

> ### Author Rebuttal · Authors · 2023-08-28
>
> Thank you for your reviews. We address your doubts and answer your questions below:
>
> **W1: The size of the test set.**
> - We test it on a new set including 400 questions on both HotpotQA and 2WikiMultihopQA (same as Trivedi et al., 2022$^1$). The performance is consistent with what we showed in the paper. The trend is the same on all semantic graph-based approaches.
> - CoT setting results on 400 random questions on HotpotQA:
>
>     | Methods         | EM     | F1     | Precision | Recall |
>     |-----------------|--------|--------|-----------|--------|
>     | Base            | 0.5925 | 0.7327 | 0.7448    | 0.7791 |
>     | G-Full prompt   | 0.6175 | 0.7634 | 0.7768    | 0.8129 |
>     | SG-Multi prompt | 0.6375 | **0.7846** | 0.7912    | **0.8489** |
>
> - CoT setting results on 400 random questions on 2WikiMultihopQA:
>
>     | Methods         | EM     | F1     | Precision | Recall |
>     |-----------------|--------|--------|-----------|--------|
>     | Base            | 0.4250 | 0.5317 | 0.5009    | 0.7238 |
>     | G-Full prompt   | 0.4625 | 0.5650 | 0.5339    | 0.7564 |
>     | SG-Multi prompt | 0.4900 | **0.6005** | 0.5728    | **0.7723** |
>
> - We sample 100 new questions in the 2WikiMultihopQA dataset and manually annotate their reasoning chains according to the format defined in Section 3.3. The result is shown in the table:
>
>     | Method          | Human | ROUGE-1 | ROUGE-2 | ROUGE-L |
>     |-----------------|-------|---------|---------|---------|
>     | Base Prompt     | 0.81  | 0.680   | 0.520   | 0.614   |
>     | G-Full prompt   | 0.87  | 0.687   | 0.536   | 0.624   |
>     | SG-Multi prompt | 0.88  | 0.689   | 0.543   | 0.628   |
>
> **W2: The significance of performance improvement and the long prompt**
> - We also focus on the faithfulness of the reasoning process, which is arguably as important as the QA performance. The prompt including the SGs (although longer) provides grounded structure information that helps improve the faithfulness of the answer and reasoning chain.
> - We do the t-test on the recall and F1 metrics on samples we select from both HotpotQA and 2WikiMultihopQA to check whether the improvement is significant. The results show that F1 and recall are significantly better (with p-value < 0.05) on the SG-Multi prompt method.
>
> **W3: The effectiveness of extracted information from LLM is a concern due to the lack of IE outcomes.**
> - We conduct the evaluation of semantic graphs based on the method in Jurafsky & Martin 2022$^2$. Currently, Semantic graphs are evaluated by humans. Human evaluators report over the 90% precision on triples in SG. -- we will add this to the final version.
>
> **Q1. Could you test the LLM in terms of IE capability? Such as evaluating some NER or relation extraction benchmarks.**
> - Our work does not rely on close-domain IE (schema-based). Thus we do test on a close domain IE banchmark (e.g. CoNLL, ACE). The human evaluation shows that our open-IE based SG is good, i.e., achieving over 90% precision (as mentioned above). We will formally report this in the final version.
>
> **Q2: The explanations generated by the method are grounded to the extracted information. How do you verify this?**
> - In our work, explanations are semantic graphs but not reasoning chains. SGs are *extracted* from documents with the *generative* approach, which makes them fully grounded to **the document context**.
> - As for the generated reasoning chains/processes, we evaluate them separately and measure their faithfulness/correctness (Figure 3).
>
>
> **References:**
> 1. Trivedi, H., Balasubramanian, N., Khot, T., & Sabharwal, A. (2022). Interleaving retrieval with chain-of-thought reasoning for knowledge-intensive multi-step questions. ACL 2023
> 2. Jurafsky & Martin 2022. Speech and Language Processing (3rd edition).

---

### Official Review · Reviewer_ke1q · 2023-08-04

**Soundness:** 4

**Excitement:**

4: Strong: This paper deepens the understanding of some phenomenon or lowers the barriers to an existing research direction.

**Paper Topic And Main Contributions:**

This work proposes a way to construct and leverage semantic graphs in the prompting setting to enhance multi-hop reasoning and improve reasoning chains. Results show that the framework introduced in this work generates more faithful reasoning chains and substantially improves the QA performance on two benchmark datasets. The extracted structures provide grounded explanations preferred by humans, as compared to the generated reasoning chains and saliency-based explanations.

**Questions For The Authors:**

A. Instead of using the fully connected graph (SG-Full) to understand the effect of the semantic graph structure, did you try using a random subgraph of this graph?

B. The fully connected graph (G-Full) does not contain relation information. Did you try dropping the relations/edges from the actual semantic graph and noting the drop in performance to understand the role of relations? In combination with A, this could tell you if the entities/nodes in the semantic graph are more important than a random set of entities.

C. The evaluations are mostly done on the reasoning chains and how grounded they are in context. Can the generated SG itself be evaluated?

D. It would be interesting to see an analysis of the error cases where the model was previously correct (w/o semantic graph), but not after. In such cases, what is the quality of the generated reasoning chain before and after?

**Reasons To Accept:**

- The paper is well-written and the claims are strongly supported.
- Producing reasoning chains that are grounded in context is essential and this work proves that by extensive human evaluation and finds that this can also improve model performance


**Reasons To Reject:**

- It is unknown if models perform reasoning in a manner that is similar to humans. In other words, a model producing more faithful reasoning might not always lead to better model performance. It is important to understand why model performance improves and whether there are spurious correlations in the data that are responsible for the improved performance.

**Reproducibility:**

3: Could reproduce the results with some difficulty. The settings of parameters are underspecified or subjectively determined; the training/evaluation data are not widely available.

**Reviewer Confidence:**

3: Pretty sure, but there's a chance I missed something. Although I have a good feel for this area in general, I did not carefully check the paper's details, e.g., the math, experimental design, or novelty.

**Typos Grammar Style And Presentation Improvements:**

- Typos: 609, 583, 498
- 534: it should be "example comparing"?
- Fig 3: typo in reasoning "chian"

---

> ### Author Rebuttal · Authors · 2023-08-28
>
> Thank you for your reviews. We address your doubts and answer your questions below:
>
> **W1: Does better reasoning faithfulness correlate with better model performance?**
> - Thanks for the good question. We focus on faithfulness, which is as important as performance. Graphs provide grounded structure information that helps improve the faithfulness of answers and reasoning chains. Based on our experiment, most of the time better reasoning faithfulness/correctness improves performance. – There is a very strong correlation (across our test examples).
>
> **Q1: How does using a random subgraph affect the performance?**
> - Random sub-graphs have a large chance of missing important triples. We have tried adding random SGs and run the method multiple times. The performance dropped dramatically.
>
> **Q2: Did you try dropping the relations/edges from the actual SG and noting the drop in performance to understand the role of relations?**
> - By dropping relations in SG-Multi, the performance drops and looks similar to the performance of G-Full. Relations help LLMs more precisely jump a hop from one entity to another entity.
>
> **Q3: Can the generated SG itself be evaluated**
> - We conduct the evaluation of semantic graphs based on the method in Jurafsky & Martin 2022$^1$. Currently, Semantic graphs are evaluated by humans. Human evaluators report over the 90% precision on triples in SG.
>
> **Q4: It would be interesting to see an analysis of the error cases where the model was previously correct (w/o semantic graph), but not after. In such cases, what is the quality of the generated reasoning chain before and after?**
> - In our observations based on the human analysis, most of the cases when adding SG leads to the error are because of the incomplete SG. – It leads to worse/unfaithful reasoning processes and incorrect answers. We will list several qualitative examples in the final version.
>
> Thank you again for the insightful suggestions, we will report the findings/results in the final version.
>
> **Reference**
> 1. Speech and Language Processing (3rd edition), Jurafsky & Martin 2022.

---

### Official Review · Reviewer_ns5s · 2023-08-05

**Soundness:** 3

**Excitement:**

4: Strong: This paper deepens the understanding of some phenomenon or lowers the barriers to an existing research direction.

**Paper Topic And Main Contributions:**

This paper focuses on using structured information for explainable multi-hop question answering and reasoning. It addresses the challenge neural models face, including large language models (LLMs), in performing multi-hop reasoning and providing well-grounded explanations. To address this issue, the paper proposes a novel framework that leverages extracted semantic structures to enhance the models' multi-hop reasoning capabilities, leading to improved task performance.
The main contributions of this work are as follows:

1) Introducing the concept of semantic graphs and their application to enhance the models' multi-hop reasoning abilities, resulting in improved task performance.

2) Demonstrating that incorporating semantic graphs in the prompting process leads to the generation of higher-quality and more accurate reasoning chains.

3) Showing that semantic graphs-based explanations contribute to context-grounded interpretations of the predicting process, which human evaluators prefer.

Empirical results and human evaluations are provided to assess the effectiveness of the proposed approach, comparing it against alternative methods in terms of generating faithful reasoning chains and delivering grounded explanations.

**Questions For The Authors:**

Q1 Despite G-Full not utilizing semantic relations (edges) and connecting all entities indiscriminately, the results are remarkably similar to SG-multi. Why is this similarity observed, given the inclusion of potentially irrelevant data? Shouldn't the performance be significantly lower due to the infusion of ineffective information?

Q2: How do the results change if we use only the graph and the question in the prompt?

Q3: How do you address coreferences, pronouns, and same mentions? For instance, when two mentions refer to the same entity, what approach do you employ to handle such cases?

Q4: Why does Table 2 only display the recall on hotpotQA? What about the results for other benchmarks and metrics to offer a more comprehensive evaluation also on the few-shot setting?

Q5: In line 372, it is mentioned something about tunning the hyperparameters, but after that, it is not mentioned why and when this tuning happened.

**Reasons To Accept:**

S1: Employing a straightforward yet highly efficient prompt engineering method to integrate data into the Language Model (LLM) during in-context learning, enhancing LLM performance in multi-hop reasoning tasks.

S2: Assessing the quality of predicted CoT following the infusion of Graph information by incorporating human judgment.

S3: Investigating the model prediction's explainability through a series of experiments involving both human evaluators and the models themselves.

**Reasons To Reject:**

W1: The text states that GPT3.5 was employed for all experiments. Although we acknowledge that each variant of GPT3.5 possesses distinct capabilities, it is crucial to specify the exact model used. For instance, InstructGPT(DaVinci-003) exhibits superior performance in reasoning and instruction-based tasks, whereas Turing and DaVinci perform comparably in extraction tasks. If ChatGPT(Turing) were utilized for reasoning, DaVinci might yield better results or vice versa. Thus, clarifying the specific GPT3.5 variant utilized can significantly impact the outcomes and conclusions drawn from the experiments.

W2: In the experiments of this paper, only the gold paragraph is utilized (line 372). However, as indicated, the size of the graph (SG-full) in certain instances becomes huge and cannot be accommodated within the prompt. Consequently, employing this method in the primary challenge, which involves considering all paragraphs, or in real-world scenarios, may prove impractical due to the substantial size of the constructed graphs, even for SG-multi and SG-one variants. This limitation may impede the applicability and feasibility of the approach in scenarios where dealing with large-scale graphs is essential.

**Reproducibility:**

3: Could reproduce the results with some difficulty. The settings of parameters are underspecified or subjectively determined; the training/evaluation data are not widely available.

**Reviewer Confidence:**

4: Quite sure. I tried to check the important points carefully. It's unlikely, though conceivable, that I missed something that should affect my ratings.

**Typos Grammar Style And Presentation Improvements:**

T1: The naming of CoT and Few-shot is confusing in this paper. While citing Wei,2022, in line 197 for CoT-setting, it's important to note that in this paper, CoT is included in the prompt. In contrast, your approach only asks the model to include CoT in the prediction. Both of your approaches (CoT-setting and few-shot setting) are actually few-shot prompting with additional graph information. To avoid confusion, consider using a different or clearer term that does not overlap with the CoT method from the original paper.

T2: Maybe an example in line 235 can make it more clear.

T3: In many cases, you have both G-Full and SG-Full (e.g., Captions and body of Table 1 and 2). Please only keep one of them.

---

> ### Author Rebuttal · Authors · 2023-08-28
>
> Thank you for your reviews. We address your doubts and answer your questions below:
>
> **W1: Different GPT3.5 variants may exhibit superior performance in different aspects.**
> - We tried different variants and there are no significant differences in terms of results. We will specify this in the final version.
>
> **W2: The size of semantic graphs is large, which impedes the applicability of the approach in scenarios where dealing with large-scale graphs is essential.**
> - The limitation of G-Full is mentioned in section 4.2. Long graphs may not fit into the LLM’s input length constraint. With high information retrieval quality, the number of relevant paragraphs will not be large. Thus, SG-Multi or SG-One will not be large and can fit into LLM’s input (>90%).
>
> **Q1: Shouldn't the performance be significantly lower due to the infusion of ineffective information?**
> - Since we use golden paragraphs, most graphs do not exceed the graph size upper limit. Reasoning chains are implicitly included. In the paper, we mentioned that some graphs of G-Full may not fit the LLM’s input size limitation. So, its performance is still slightly above baseline and is lower than the other two SG-based methods.
>
> **Q2: How do the results change if we only include the graph and the question in the prompt?**
> - If we only input graphs and questions, some details from the original text are ignored (e.g. one exact day). It leads to a worse performance. Documents contain details that graphs may not cover, which may affect the results.
>
> **Q3: How do you address coreferences, pronouns, and same mentions?**
> - According to our human evaluation and analysis on semantic graphs, co-referential entities can be implicitly recognized by LLMs (often with help from the SGs).
>
> **Q4: Why does Table 2 only display the recall on HotpotQA?**
> - Recall is the most important metric among all four metrics. We have discussed it in section 4.2. In this table, we show that the semantic graph-building method brings the most improvement in recall. Other metrics also describe the same trend as recall. However, their improvements are not as significant as on the recall metric.
>
> **Q5: In line 372, it is mentioned something about tunning the hyperparameters, but after that, it is not mentioned why and when this tuning happened.**
> - We follow the settings in the paper Trivedi et al., 2023$^1$. The parameters for generation are required by OpenAI API, such as temperature, top_p for sampling, etc. Most parameters are default values. The temperature for generation is set to 0 for diversity control.
>
> **Reference**:
> 1. Trivedi, H., Balasubramanian, N., Khot, T., & Sabharwal, A. (2022). Interleaving retrieval with chain-of-thought reasoning for knowledge-intensive multi-step questions. ACL 2023

---

### Meta-Review · Senior_Area_Chairs · 2023-10-05

**Recommendation:** 4

**Metareview:**

The work suggests to additionally feed semantic graphs that represent input documents as part of the prompt when answering multi-hop questions. The authors show that on two multi-hop QA datasets, feeding the semantic graphs helps the model to predict more faithful reasoning chains and more accurate answers. The work is sound as agreed by all reviewers (3,4,3) and relatively exciting (4,4,3). I also find it appealing as it introduces a simple way to use structure in LLMs with CoT reasoning, that shows some empirical benefit. Given the above, I think it should be accepted to the main conference or Findings.

---

### Decision · Program_Chairs · 2023-10-07

**Decision:**

Accept-Findings

**Comment:**

The work suggests to additionally feed semantic graphs that represent input documents as part of the prompt when answering multi-hop questions. The authors show that on two multi-hop QA datasets, feeding the semantic graphs helps the model to predict more faithful reasoning chains and more accurate answers. The work is sound as agreed by all reviewers (3,4,3) and relatively exciting (4,4,3). I also find it appealing as it introduces a simple way to use structure in LLMs with CoT reasoning, that shows some empirical benefit. Given the above, I think it should be accepted to the main conference or Findings.